Effects of combined immunosuppressant and hepatitis B virus antiviral use on COVID-19 vaccination in recipients of living donor liver transplantation

Lee Ryunjin 1
http://orcid.org/0000-0002-4957-3219 Choi Jiwan 1
Lee Eunkyeong 1
http://orcid.org/0000-0002-7938-8196 Lee Jooyoung 1
http://orcid.org/0000-0002-5758-6810 Kim Jiye 1
Kang Seoon 1
An Hye-In 1
Kim Sung-Han 2
Kim Sung-Min 3
Jwa Eun-Kyoung 3
Park Gil-Chun 3
Namgoong Jung-Man 4
Song Gi-Won 3
Yoon Young-In 3 ltsyoon@amc.seoul.kr
Tak Eunyoung 1 eunyoung.tak@amc.seoul.kr
Lee Sung-Gyu 3
1 Department of Convergence Medicine, Asan Medical Institute of Convergence Science and Technology (AMIST) , Seoul , Republic of South Korea
2 Department of Infectious Diseases, Asan Medical Center, University of Ulsan College of Medicine , Seoul , Republic of South Korea
3 Division of Hepatobiliary Surgery and Liver Transplantation , Seoul , Republic of South Korea
4 Division of Pediatric Surgery, Department of Surgery, Asan Medical Center, University Ulsan College of Medicine , Seoul , Republic of South Korea
Verma Sheetal
Electronic publication date: 2024 Dec 6
Publication date: 2024
Volume: 12
Electronic Location ID: e18651
Received 2024 Jun 19; Accepted 2024 Nov 15
Copyright: © 2024 Lee et al.
Copyright year: 2024
Copyright holder: Lee et al.
License: This is an open access article distributed under the terms of the Creative Commons Attribution License, which permits unrestricted use, distribution, reproduction and adaptation in any medium and for any purpose provided that it is properly attributed. For attribution, the original author(s), title, publication source (PeerJ) and either DOI or URL of the article must be cited.
License URL: https://creativecommons.org/licenses/by/4.0/

Keywords: Hepatitis B virus, Living-donor liver transplantation, Anti-viral drug, COVID-19

Funding: National Research Foundation of Korea NRF-2022R1A2C200614111 and NRF-2015K1A4A3046807 Asan Institute for Life Sciences, Asan Medical Center, Seoul, Korea AMC- 2021IL0010 Funding was provided by the National Research Foundation of Korea (grant numbers NRF-2022R1A2C200614111 and NRF-2015K1A4A3046807) and the Asan Institute for Life Sciences, Asan Medical Center, Seoul, Korea (grant number AMC- 2021IL0010). The funders had no role in study design, data collection and analysis, decision to publish, or preparation of the manuscript.

==============================
Background & Aims

The global pandemic caused by the highly contagious SARS-CoV-2 virus led to the emergency approval of COVID-19 vaccines to reduce rising morbidity and mortality. However, limited research exists on evaluating the impact of these vaccines on immunocompromised individuals, such as recipients of living donor liver transplantation, highlighting the need for further studies to better understand their effectiveness in this specific population.

Methods

From June 2021, we followed up on the effectiveness of the vaccine for patients taking immunosuppressive drugs after living-donor liver transplantation (LDLT). A total of 105 immunocompromised individuals participated, of which 50 patients with hepatitis B were taking antiviral drugs. Patients were assessed to analyze how the combination of immunosuppressive and antiviral drugs affected the efficacy of the BNT162b2, mRNA-1273, and ChAdOx1 nCoV-19 COVID-19 vaccines.

Results

Before and after the vaccinations, patients were monitored to establish differences between immunosuppressed patients and those additionally taking antiviral drugs. In immunocompromised patients taking antiviral drugs for hepatitis B, we confirmed that the effect of the COVID-19 vaccine was reduced when compared to immunocompromised patients. Interestingly, 23 patients (11 without and 12 additionally with hepatitis B drug administration) encountered breakthrough infections, and although there was a minor discrepancy in vaccine efficacy among the patients taking antiviral drugs for hepatitis B, it did not reach statistical significance.

Conclusions

Additional COVID-19 vaccination is recommended for patients taking immunosuppressive drugs and hepatitis B antiviral drugs after LDLT.

Introduction

In December 2019, SARS-CoV-2 (type 2 severe acute respiratory syndrome coronavirus, also known as COVID-19), a highly contagious virus that causes systemic symptoms with severe respiratory disease, quickly spread worldwide. The World Health Organization (WHO) subsequently declared it a pandemic in March 2020 (Ciotti et al., 2022; Villapol, 2020; Mahajan et al., 2021). Vaccines like mRNA (BNT1621b2, and mRNA-1273) and adenovirus-vectored (AZD1222) were developed, with the Food and Drug Administration (FDA) approving vaccines BNT1621b2 and mRNA-1273 on December 11 and 18, 2020, respectively (Folegatti et al., 2020; Kim et al., 2021a; Ramasamy et al., 2021; Graham, 2020). On February 26, 2021, COVID-19 vaccination commenced in South Korea, utilizing the mRNA vaccines (BNT1621b2 and mRNA-1273) and the ChAdOx1 nCoV-19 vaccine. In light of government-approved vaccines, researchers initiated a study to investigate the effects of these three authorized vaccines on immunocompromised individuals with hepatitis B.

Symptoms of COVID-19 may range from mild to severe, including lung damage and death. Solid organ transplant recipients, due to immunosuppression, are more susceptible to COVID-19. Immunocompromised patients experience similar severe infection rates but possess fewer active immune cells (Struyf et al., 2022; DeWolf et al., 2022; Banerjee et al., 2020; Di Maira, Little & Berenguer, 2020; Rovira, Mascarell & Bachi, 2000; Eckerle et al., 2013; Rabinowich et al., 2021; Thuluvath, Robarts & Chauhan, 2021; Timmermann et al., 2021; Hall et al., 2021; Hall, Humar & Kumar, 2022; Samidoust et al., 2021; Zaky et al., 2021; Ruether et al., 2022). Moreover, some transplant recipients are administered antiviral drugs, e.g., those patients suffering from the hepatitis B virus (HBV), both before and after transplantation. However, with the recent development of antiviral drugs such as tenofovir disoproxil (Viread/TDF), tenofovir alafenamide (Vemlidy/TAF) and entecavir (Baraclude), the recurrence of HBV infection has been reduced (Samuel et al., 1993; Todo et al., 1991). Antiviral drugs are known to target the pathogenicity of viruses or compete with virions for binding sites on the surface of host cells (Kausar et al., 2021). Thus, they can affect host immunity and inhibit immune function (Heagy et al., 1991).

Therefore, we hypothesized that antiviral treatment of LDLT recipients may adversely impact the efficacy of COVID-19 vaccines. During antiviral therapy, the virus can adapt to the drug and develop escape variants. This can weaken the effectiveness of the antiviral treatment and lead to resistance to vaccines or other treatment methods. However, research on the efficacy of combined antiviral therapy and vaccines is lacking (Sheldon & Soriano, 2008). Here, we measured antibody levels to assess the COVID-19 vaccine’s impact on liver transplant-related medications.

Patients and methods

Information on participants and samples

All patients (n = 105) participating in this study were vaccinated against COVID-19 with the BNT162b2 (n = 60), mRNA-1273 (n = 11), and ChAdOx1 nCoV-19 (n = 16) vaccines after receiving liver transplantation. Patients aged 18 or older without a documented history of SARS-CoV-2 infection were only included in the study if they had a baseline SARS-CoV-2 IgG negative result. In the final analysis, only 87 recipients were included; 18 recipients were excluded for the following reasons: withdrawn consent (n = 8), dropout (n = 9), and discontinued treatment (n = 1). Participants had no previous diagnosed (SARS-CoV2 IgG negative results) SARS-CoV2 infection before vaccination. Immunocompromised patients, including those with HBV, received three doses of COVID-19 vaccines according to specific schedules. Both BNT162b2 and mRNA-1273 were administered at 0, 1, 2, 3, and 6 months, while ChAdOx1 nCoV-19 was administered at 0, 1, 2, and 6 months for general immunocompromised patients, and at 0, 1, 2, 3, and 6 months for those with HBV.

Breakthrough infections were defined as SARS-CoV-2 infections in partially or fully vaccinated participants that had never been infected with SARS-CoV-2. The details of participant selection are described in Fig. 1. The general healthy control group without liver transplantation consisted of healthcare workers (n = 134). COVID-19 vaccines were prioritized for vulnerable populations, including individuals with immunodeficiency and healthcare workers. Both the healthy control group and immunocompromised individuals received early vaccinations. Among those who received the BNT162b2 vaccine (n = 34) in the healthy control group of 134 individuals, the average age was 32.0 (range 23–64) years, while those who received the mRNA-1273 vaccine (n = 16) had an average age of 26.5 (range 24–53) years. Additionally, the healthy control group receiving the ChAdOx1 nCoV-19 vaccine (n = 84) had an average age of 36.0 (range 21–64) years. A significant age difference was observed among the healthy control groups (p < 0.001). Furthermore, it is noteworthy that the healthy control group had no previously diagnosed cases of SARS-CoV-2 infection. Although participants completed the 1st, 2nd, and 3rd vaccination doses, data from blood samples taken 6 months post-vaccination are not available. The COVID-19 vaccination schedule varied in terms of the interval between the 1st and 2nd doses, depending on the vaccine type. According to the manufacturer’s recommendations, the BNT162b2 and mRNA-1273 vaccines were advised to be administered with an interval of 8 to 12 weeks, while the ChAdOx1 nCoV-19 vaccine was recommended to be administered with an interval of 4 to 12 weeks. Researchers made efforts to collect blood samples in accordance with the respective manufacturer’s recommended intervals post-vaccination. This study was approved by the Institutional Review Board of the Asan Medical Center (authorization no. 2021-0746). All volunteers provided written informed consent. The research was conducted in accordance with the Declaration of Helsinki.

Figure 1 Study design flow diagram of SARS-CoV-2 vaccinated patients following liver transplantation.

Preparation of plasma and measurement of the antibody response

In all cases, 10 mL of peripheral blood in lithium heparin solution was collected from each individual, along with an Ethylenediaminetetraacetic acid (EDTA)-containing sample from the same individuals. Plasma was obtained by centrifuging at 1,200 × g for 20 min at room temperature with the brake off, using Lymphoprep (Cat # 07861; STEMCELL Technologies, Vancouver, Canada). Subsequently, the obtained plasma was stored at −40 °C. SARS-CoV S1-specific IgG Ab titers were measured using an enzyme-linked immunosorbent assay (ELISA) qualified with reference pooled sera from the International Vaccine Institute (Seoul, Korea). S1-specific IgG Ab titers are presented in international units per milliliter. To determine the cutoff values for the ELISA, the mean and SD (standard deviation value) of the negative control plasma were measured, and cutoff values were defined as the mean plus three-fold the SD value. The cutoff value was 10 IU/mL for IgG (Lim et al., 2022; Kim et al., 2022).

Assessment indicators of liver abnormalities in the patients

COVID-19 is a recently emerged infectious disease, and there is currently no indicator to determine liver damage caused specifically by it. Moreover, the elevated enzyme levels in liver tests ranged from mild to moderate. Thus, we defined the pattern of liver abnormalities in terms of elevation of the following liver enzymes in the serum: ALT (alanine aminotransferase) >40 U/L; AST (Aspartate aminotransferase) >40 U/L; and TBIL (total bilirubin) 0.2–1.2 mg/dL.

Statistical analysis

All statistical analyses and graph plotting were performed using GraphPad Prism 8.0 software (GraphPad Software, San Diego, CA, USA). Statistical analysis was performed using the Wilcoxon matched-pairs signed rank test when comparing two groups, and one-way ANOVA was used for statistical analysis when comparing multiple groups. Logistic and linear regression models were fitted to determine the factors influencing the seropositivity and Ab titer after vaccination. A two-tailed p-value of <0.05 was considered statistically significant in all analyses.

Results

Participant characteristics

In our study, a total of 105 patients participated, but only 87 patients were included in the analysis (Fig. 1). Additionally, the baseline characteristics were assessed for 134 healthy control group individuals that received the adenoviral vector vaccine (BNT162b2, n = 34, 25%), mRNA vaccine (mRNA-1273, n = 16, 12%), adenoviral vector vaccine (ChAdOx1 nCoV-19, n = 84, 63%) enrolled in this study. All 87 patients that participated in the study were administered immunosuppressive drugs after LDLT. Among them, 50 patients (57.5%) took immunosuppressant and HBV antiviral drugs in combination (some discontinued HBV antiviral drugs after LDLT, n = 10). The average age of this group was 58 years, and most were male (70%). All participants were vaccinated in June 2022 with the BNT162b2 (n = 34, 68%), mRNA vaccine (mRNA-1273, n = 3, 6%), or adenoviral vector vaccine (ChAdOx1 nCoV-19, n = 13, 26%). In comparison, 37 patients (42.5%) received only immunosuppressant after liver transplantation. These participants were vaccinated with the BNT162b2 (n = 27, 72.9%), mRNA-1273 (n = 7, 18.9%), and ChAdOx1 nCoV-19 (n = 3, 8.1%) vaccines. All patients were vaccinated up to the 2nd dose, but there were a few immunocompromised patients (n = 4) and hepatitis B immunocompromised patients (n = 3) that refused the 3rd vaccine. Immunocompromised hepatitis B patients took the antiviral drugs Entercavir (BaracludeR) and Tenofovir (Viread) (Table 1).

Table 1 Baseline characteristics of immunocompromised and HBV patients.

	Vaccinated immunocompromised (n = 37)	Vaccinated HBV (n = 50)	p-value	
Sex				
Male	21	35		
Female	16	15		
Age group				
Median (min–max)	49 (22–63)	58 (45–74)	<0.0001	
18–29	2	0		
30–39	5	0		
40–49	13	7		
50–59	12	30		
60–64	5	9		
65+	0	6		
Initial vaccination type				
BNT162b2	27	34		
mRNA-1273	7	3		
ChAdOx1 nCoV-19	3	13		
Cross-vaccinated	3	11		
Number of doses				
1	37	50		
2	37	50		
3+	33	47		
Anti-hepatitis B virus drugs				
Entercavir (BaracludeR)		20		
Tenoforvir (Viread)		20		
Immunosuppressants				
Tacrolimus (FK506)	11	15		
Mycophenolate or mofetil	1	5		
Tacrolimus + Mycophenolate or mofetil	18	17		
Tacrolimus + Everolimus	7	13		

The concentration of SARS-CoV-2 S1-specific IgG antibodies in the blood after vaccination is a crucial indicator for assessing vaccine efficacy. These specific antibodies play a vital role in the immune response induced by vaccination. Upon vaccination, the immune system recognizes viral components, such as the S1 protein of SARS-CoV-2 and generates specific antibodies. These antibodies circulate in the bloodstream, neutralizing and eliminating the virus following subsequent exposure. Therefore, measuring the concentration of SARS-CoV-2 S1-specific IgG antibodies in the blood of vaccinated individuals allows researchers to evaluate the strength and persistence of the vaccine-induced immune response. To assess the vaccination effect on each group, we measured a SARS-CoV-2 S1-specific antibody in the patients’ blood that was collected at six intervals.

We conducted multiple comparisons to analyze the results. Levels of the SARS-CoV-2 S1-specific antibody were significantly higher in the healthy control group compared to the immunocompromised patients (p < 0.0001). Furthermore, antibody levels were significantly lower in immunocompromised patients receiving anti-HBV treatment than in the healthy control group (p < 0.0001). Among immunocompromised patients, there was also a significant difference observed, with antibody levels increasing with an increasing number of vaccinations (p = 0.0202) (Fig. 2).

Figure 2 Response of S1-specific IgG antibody titers after the first, second, and third dose of COVID-19 vaccine with LT patients and healthy control group (excluding breakthrough).

Kinetics of S1-IgG titers induced by COVID-19 vaccine in the healthcare worker control, immunocompromised and immunocompromised with HBV antiviral drug user groups. The participants with breakthrough COVID-19 infections were excluded from the graph (healthy control n = 134, immunocompromised n = 37, immunocompromised with anti-HBV n = 50; 1st dose, first vaccine dose; 2nd dose, second vaccine dose; 3rd dose, third vaccine dose; 6 Mo, 6 months after vaccination). One-way ANOVA (Tukey’s multiple comparisons test).

When comparing the antibody levels between vaccinated individuals and healthy controls, there was no significant difference observed before vaccination or after the 1st dose (p = 0.9436). However, as the number of vaccine doses increased, antibody levels also increased. In particular, the most significant differences were observed between before vaccination and after the 3rd dose, as well as between after the 1st and 3rd doses (p < 0.0001). Significant differences were also noted for before vaccination and after the 2nd dose (p = 0.0002). Additionally, antibody levels increased after the 1st and 2nd doses (p = 0.0021). These trends indicate that antibody levels progressively increase with vaccination, and that the increase becomes more pronounced with additional doses. Furthermore, when comparing immunocompromised patients excluding healthy controls, no significant differences were observed between immunocompromised patients and immunocompromised hepatitis B patients before the first and second vaccination doses. However, considerable differences in antibody levels were observed between immunocompromised patients and healthy controls. Nevertheless, immunocompromised patients showed a gradual increase in antibody levels with each dose. Significant results were observed for before vaccination and after the 3rd dose (p = 0.0014). Furthermore, a noteworthy difference was observed after 6 months post-vaccination (p < 0.0001). This indicates that even in immunocompromised patients, consistent vaccination leads to antibody generation, significantly differentiating them from their pre-vaccination status and indicating the increased efficacy of the vaccine (Figs. S1 and S2).

Immune response and liver function tests according to the vaccine type

Immunodeficiency patients and hepatitis B immunodeficiency patients that underwent LDLT were found to differ markedly in their immune responses as they were vaccinated. Of a total of 37 patients taking immunocompromised drugs after LDLT, 72.9% (n = 27) were given the BNT162b2 vaccine, 18.9% (n = 7) received mRNA vaccines (mRNA-1273), and 8.1% (n = 3) were given adenoviral vector vaccine (ChAdOx1 nCoV-19).

Among 50 immunocompromised hepatitis B patients, 68% (n = 34) were given BNT162b2 vaccine, 6% (n = 3) received the mRNA-1273 vaccine, and 26% (n = 13) were administered the ChAdOx1 nCoV-19) vaccine. In both groups, the majority of patients received the BNT162b2 vaccine (Fig. S3). In addition, we conducted liver function tests before and after vaccination on the patients that participated in the study and compared them. Liver function tests measure the ALT, AST, and TBIL levels. Our analysis revealed no difference in liver function following LDLT before and after vaccination (p > 0.05). Therefore, concerns about liver function deterioration caused by vaccines can be safely disregarded (Fig. 3).

Figure 3 Liver function test results before and after vaccination in vaccinated immunocompromised and vaccinated immunocompromised anti-HBV patients.

Vaccinated immunocompromised patients (n = 37); before vaccination (ALT 14 (5–63), AST 23 (15–56), TBIL 0.6 (0.3–2.5), albumin 4 (2.7–4.5)), after vaccination (ALT 14 (5–65), AST 22 (11–135), TBIL 0.6 (0.3–1.9), albumin 3.9 (2.4–4.6)). Vaccinated immunocompromised anti–HBV (n = 50); before vaccination (ALT 18 (5–77), AST 23.5 (13–167), TBIL 0.65 (0.2–11.4), albumin 4 (2.8–4.4)), after vaccination (ALT 17 (5–103), AST 25 (9–140), TBIL 0.7 (0.3–4.2), albumin 4 (2.6–4.6)). Values are presented as median range (minimum range, maximum range); ALT, alanine aminotransferase (normal range < 40 IU/L); AST, aspartate aminotransferase (normal range < 40 IU/L); TBIL, total bilirubin (normal range 0.2–1.2 mg/dL); albumin (normal range 3.5–5.2 g/dL).

Breakthrough infections in vaccinated participants receiving immunosuppressants and HBV antiviral drug combinations

Among immunocompromised patients receiving immunosuppressive drugs after LDLT and hepatitis B immunosuppressed patients, tacrolimus (FK506) and mycophenolate (mofetil) were commonly used (Song et al., 2016; Jeng et al., 2018; Namgoong et al., 2013). Among the immunocompromised patients taking immunosuppressive drugs after LDLT, 29.73% (n = 11) of patients took tacrolimus (FK506) alone, and among the hepatitis B immunosuppressed patients, this was 30% (n = 15). A minimal number of patients (immunocompromised patients that took immunosuppressive drugs after LDLT; 2.70%, n = 1) and hepatitis B immunocompromised patients (10%, n = 5) received mycophenolate mofetil. Patients taking tacrolimus (FK506) and everolimus were immunosuppressed in 18.92% (n = 7) and hepatitis B immunosuppressed in 26% (n = 13) of cases, respectively. Moreover, there was no statistically significant difference observed in the SARS-CoV-2 S1-specific IgG antibody titer response between the administration of entecavir and tenofovir (Figs. 4A–4C).

Figure 4 Types and proportions of immunosuppressants and liver function tests (AST, ALT, albumin, PT (%), PT (INR), total bilirubin) on breakthrough infection patients.

(A) Types of immunosuppressive drugs taken by immunocompromised people. (B) Types of immunosuppressive drugs taken by immunocompromised patients with hepatitis B. (C) Response of SARS-CoV-2 S1-specific IgG antibody titers in patients with HBV taking entecavir and tenofovir (entecavir; n = 20, tenofovir; n = 20, p = 0.4688 n.s.). (D) Liver function tests were conducted before and after breakthrough infections in two groups: one with a general breakthrough infection (n = 11) and another with breakthrough infection involving hepatitis B (HBV) (n = 12). PT (%), % Prothrombin time; PT (INR); prothrombin time international normalized ratio.

In the immunocompromised group with hepatitis B, the antiviral drugs entecavir (BaracludeR) or tenofovir (Viread) were administered (Yoon et al., 2015). Entecavir is rapidly converted in vivo to its active form, entecavir-triphosphate, and inhibits three steps of virus replication: priming of HBV DNA polymerase, reverse transcription of HBV DNA negative strands from pre-genomic mRNA, and synthesis of HBV DNA positive strands (Scott & Keating, 2009). Tenofovir is a competitive inhibitor of HBV polymerase, functioning as a nucleotide analog that inhibits viral replication by competing with natural nucleotides for binding to the active site of HBV polymerase (De Clercq, 2016). Among the 20 patients taking entecavir (BaracludeR) and the 20 taking tenofovir (Viread) the male to female ratio (male = 6 and female = 14) was consistent and had similar age (median age = 59). Vaccination rates were also consistent, with both groups taking antivirals up to the 3rd vaccine. The most commonly administered vaccine was BNT162b2.

After the first inoculation, there was little difference in the median antibody titer between the two antiviral drug groups: entecavir = 2.82 IU/mL and tenofovir = 3.55 IU/mL. However, 3 weeks after the second vaccination, the median antibody titer of tenofovir (53.37 IU/mL) was significantly higher than that of the entecavir (36.79 IU/mL) group. In the blood results collected after 2 weeks, we confirmed that the difference in antibody value between the entecavir (284 IU/mL) and tenofovir (303.09 IU/mL) groups was not significant. However, after the third inoculation, there was a considerable difference, with the antibody value of the tenofovir (1,064.50 IU/mL) group rising markedly higher than that of the entecavir (269.45 IU/mL) group. The patient with the highest antibody value was observed in the entecavir (3,097.75 IU/mL) group, and in the tenofovir (5,792.88 IU/mL). Among the immunocompromised hepatitis B patients taking immunosuppressive drugs, the vaccine antibody value was found to be much higher in the patients taking tenofovir than those taking entecavir (Table 2).

Table 2 Type of antiviral drugs.

	Anti-hepatitis B virus drugs
(Entercavir (BaracludeR) = 20)	Anti-hepatitis B virus drugs
(Tenoforvir (Viread) = 20)	p-value	
Sex				
Male	6	6		
Female	14	14		
Age group				
Median (min–max)	59 (47–66)	59 (45–74)	0.956945	
40–49	2	3		
50–59	11	10		
60–64	5	4		
65+	2	3		
Initial vaccination type				
BNT162b2 (pf)	12	13	0.1642	
mRNA-1273 (m)	2	1	0.3632	
ChAdOx1 nCoV-19 (az)	6	6	0.7180	
Cross-vaccinated	5	5		
Number of doses				
1	20	20		
2	20	20		
3	20	20		
4	7	9		
Note:

As a result of the investigation, 10 people were excluded because they were not taking antiviral drugs.

Breakthrough infections occurred during the study, with a total of 23 patients exhibiting infection. After LDLT, 11 immunocompromised patients taking immunosuppressive drugs and 12 with hepatitis B had breakthrough infections. Most of these patients were infected after the second COVID-19 vaccination, with the proportion of breakthrough cases that had been vaccinated with BNT162b2 being relatively high. However, when comparing the liver function test results (AST, ALT, albumin, PT (%) (the prothrombin time, expressed as a percentage, which measures the time it takes for blood to clot), PT (INR) (the International Normalized Ratio, indicating the international standardization of prothrombin time), and TBIL) before and after breakthrough infections between the two groups, there were no significant differences observed (Figs. 4D, S4).

Notably, three of the HBV drug-taking immunocompromised patients were confirmed to have been vaccinated four times before the breakthrough COVID-19 infection. Between the two groups, there was no significant difference in gender and the vaccine type, but a statistically significant difference in age was found (p = 0.0278) (Table 3).

Table 3 Patients for breakthrough COVID-19 infection.

	Breakthrough COVID immunocompromised (n = 11)	Breakthrough COVID immunocompromised anti-HBV
(n = 12)	p-value	
Sex				
Male	7	7		
Female	4	5		
Age group				
Median (min–max)	55 (45–66)	49 (22–63)	0.027775	
18–29	1	0		
30–39	0	0		
40–49	5	2		
50–59	4	7		
60–64	1	2		
65+	0	1		
Initial vaccination type				
BNT162b2	9	7		
mRNA-1273	1	3		
ChAdOx1 nCoV-19	1	2		
Cross-vaccinated	2	2		
Number of doses				
1	11	12		
2	11	12		
3	10	10		
4	0	3		

Discussion

Several studies have now reported comparative analyses of the reactogenicity and immunogenicity of COVID-19 vaccines, employing systematic review and meta-analysis approaches. In contrast, there are few studies directly comparing the immunogenicity of adenoviral vector-based vaccines, such as ChAdOx1 nCoV-19, and mRNA-based vaccines, including BNT162b2 and mRNA-1273, against COVID-19. Furthermore, research efforts are currently underway among numerous investigators to explore the effectiveness of these vaccines in immunocompromised patients.

Among the participants of this study, immunosuppressed subjects with hepatitis B showed no significant difference in the vaccine antibody value after the first vaccination. However, the antibody response after the second vaccination was consistently lower in patients taking immunosuppressants in combination with anti-HBV drugs. This suggests that the antibody response may be lower in those taking antiviral drugs than in those who are not. Therefore, whether or not a patient is taking antiviral drugs should be considered when preparing a vaccination plan.

Nevertheless, the antibody response of both immunocompromised and immunocompromised anti-HBV groups consistently increased after the second and third additional vaccinations, since there was no difference in liver function before and after vaccination. While liver disease-related biomarkers were not available throughout the entire follow-up period, there was no significant difference in liver function between these two groups before and after vaccination. The antibody response data showed that additional vaccinations are strongly recommended for immunosuppressed patients taking immunosuppressive drugs after LDLT, even if they were taking anti-HBV drugs. This was confirmed by the progressive increase in antibody levels with an increase in vaccine doses. Furthermore, as time passed, the number of patients with breakthrough infections gradually increased. It is recommended to administer a booster dose before the expected seasonal increase in COVID-19 levels in the community to reduce the risk of infection.

Since the introduction of vaccines targeting the SARS-CoV-2 virus, especially the mRNA platforms during the COVID-19 pandemic, factors that can interact with the vaccine response or hinder its efficacy are not well understood. In particular, several studies have reported that the vaccination outcome in patients living with human immunodeficiency virus (PLWHIV) was poorer compared to non-HIV infected individuals (Huang et al., 2022; Levy et al., 2021; Spinelli et al., 2021; Bessen et al., 2022). Considering that PLWHIV shares immunological similarities with liver transplant recipients with chronic HBV, extensive investigation is required to establish effective vaccination plans for immunocompromised individuals. According to the Korean Network for Organ Sharing (KONOS), HBV was the second most prevalent factor necessitating liver transplantation (18.43%), and about 55.9% of patients that underwent liver transplantation in South Korea in 2019 had HBV (Kim et al., 2021b; Thio et al., 2002; Konopnicki et al., 2005; Bellini et al., 2009). Our research team has previously investigated the decrease in vaccine efficacy in patients taking immunosuppressants after liver transplantation (Lim et al., 2022). Considering that the mRNA vaccine can induce immune responses similar to those of viral infection in mice (Lu et al., 2020), it is feasible that mRNA vaccines may interact with several antiviral drugs. However, in a multicenter investigation of the immunogenicity of SARS-CoV-2 vaccines in patients with chronic liver diseases, HBV infection did not affect the outcome of antibody responses after vaccination (Ai et al., 2022).

As no previous studies have investigated liver transplant recipients that take antiviral drugs along with immunosuppressive drugs, we provided the first study addressing this gap in the literature. As liver damage is often fatal for patients that have undergone liver transplantation, vaccination should be avoided if it can cause any liver injury. In our investigation, none of the vaccines administered were found to increase any liver injury-related indicators. The concurrent use of immunosuppressants and antiviral drugs lowered antibody responses after SARS-CoV-2 vaccinations. However, subsequent vaccinations increased the antibody levels. Interestingly, one study reported that patients with chronic HBV had a relatively low risk of SARS-CoV-2 infection. Indeed, even after infection, the use of antiviral drugs did not change the severity of their symptoms (Kang et al., 2021). Furthermore, another study reported that patients with chronic HBV experienced rapid recovery after SARS-CoV-2 infection, and the use of TDF was considered a significant factor contributing to their rapid recovery (Xiang & Zheng, 2021). As antiviral drugs such as TDF can reportedly reduce viral load burden in SARS-CoV-2 infections (Parienti et al., 2021), their continued use may exert a negative effect on the antibody response after vaccinations, but may be conducive to recovery after SARS-CoV-2 infection.

Naturally, our study was not without limitations. First of all, the number of vaccine groups for immunocompromised patients taking immunosuppressive drugs after LDLT and immunocompromised patients with hepatitis B was very small, and the proportion of patients taking each vaccine was dissimilar. Additionally, study limitations include small sample sizes, the impact of vaccine types, and the age group bias due to the governmental vaccine policy. The age of participants that received BNT162b2 was younger than that of participants who received ChAdOx1 nCoV-19, and the mRNA-1273 vaccine was older than BNT162b2 due to government policy. In addition, data on the Alpha, Beta, Gamma, Delta, and Omicron variants of COVID-19 were not available at the time of the study. Another major limitation is the lack of measurement of various T cell cytokines. Moreover, an investigation into cellular immune responses was not conducted. We did not investigate the different antibody responses observed between vaccines. One possible explanation could be the difference in spike protein structure between BNT162b2 without a proline mutation and ChAdOx1 containing two proline mutations that induce Ab responses targeting various parts of the spike protein, especially the receptor-binding domain (Walsh et al., 2020; Stebbings et al., 2022). Another explanation could be the platform differences between adenoviral vectors (Jacob-Dolan & Barouch, 2022). This is an area that will require further research in the future.

Despite these limitations, our study provides valuable data on the differences in the dynamics of the immune response between immunocompromised individuals taking immunosuppressive drugs and those with hepatitis B that receive antiviral drugs after LDLT. In addition, it can provide guidelines for the treatment of patients with hepatitis B, high-risk groups preparing for LDLT, solid organ transplantation, or patients just before surgery. As the immune response can be confirmed according to the type of vaccine and the number of inoculations, it is possible to provide an accurate basis to decide whether additional vaccination is necessary for immunocompromised patients with hepatitis B. Furthermore, recommendations for additional vaccinations can benefit from the insights revealed by our results. In future follow-up studies, the causal relationship between immunosuppressive drugs, antiviral drugs, and vaccines should be investigated.

Conclusion

This study assessed COVID-19 vaccine responses in immunocompromised individuals, including those with hepatitis B. In recipients of LDLT that are taking immunosuppressants, the efficacy of the COVID-19 vaccine was found to be lower compared to the healthy control group. However, HBV-positive individuals exhibited suboptimal responses, and our understanding of the mRNA vaccine’s effectiveness remains limited. The influence of antiviral drugs on vaccine responses requires further investigation. Notably, the efficacy of the SARS-CoV-2 vaccine is reduced in patients taking immunosuppressants and antiviral agents concurrently, possibly due to the impact of antiviral medication on post-vaccine antibodies.

Supplemental Information

Supplemental Information 1 Response of S1-specific IgG antibody titers after the 1st, 2nd and 3rd dose of COVID-19 vaccine with LT patients (excluding breakthrough).

Kinetics of S1-IgG titers induced by COVID-19 vaccine in immunocompromised and immunocompromised with HBV antiviral drug users. Participants with breakthrough COVID-19 infections were excluded from the graph. (Immunocompromised n = 37, Immunocompromised with anti-HBV n = 50, (Wilcoxon matched-pairs signed rank test p = 0.0313). The statistics presented in the Figure represent the outcomes of the Tukey’s multiple comparisons test. All legends not indicated are denoted as n.s.; 1st 3W, 3 weeks after the first dose of vaccination; 2nd pre, Before the second dose of vaccination; 2nd 3W, 3 weeks after the second dose of vaccination; 2nd 5W, 5 weeks after the second dose of vaccination; 3rd, Third dose of vaccination; 6M, 6 months after vaccination.

Supplemental Information 2 Receiver operating characteristic (ROC) curves of the COVID-19 vaccine LT patients.

Immunocompromised and immunocompromised with anti-HBV patients.

Supplemental Information 3 Antibody responses with the ChAdOx1 nCoV-19, mRNA-1273 and BNT162b2 vaccines immunocompromised and HBV patients.

Kinetics of S1-IgG titers induced by CoV-19 vaccine from baseline to 6 month after the 1st vaccination. (A) BNT162b2 vaccine for immunocompromised patients taking immunosuppressants following living liver transplantation (PF). (B) mRNA-1273 vaccine (M). (C) adenoviral vector vaccine (ChAdOx1 nCoV-19) (AZ). (D) BNT162b2 vaccine for HBV immunocompromised patients (PF). (E) mRNA-1273 vaccine (M). (F) adenoviral vector vaccine (ChAdOx1 nCoV-19) (AZ).

Supplemental Information 4 Receiver operating characteristic (ROC) curve showing the liver function tests in LT patients before and after COVID-19 breakthrough.

Supplemental Information 5 Raw data.

Abbreviations

HBV Hepatitis B virus

ALT Alanine aminotransferase

AST Aspartate aminotransferase

BNT162b2 BioNTech/Pfizer BNT162b2 messenger RNA

BNT162b2 vaccine BioNTech/Pfizer BNT162b2 messenger RNA (Pfizer)

ChAdOx1 nCoV-19 ChAdOx1 nCoV-19 vaccine (AZD1222)

ChAdOx1 nCoV-19 vaccine ChAdOx1 nCoV-19 vaccine AZD1222 (AstraZeneca)

COVID-19 SARS-CoV-2 (Type 2 Severe Acute Respiratory Syndrome Coronavirus)

ELISA Enzyme-linked immunosorbent assay

LDLT living-donor liver transplantation

mRNA-1273 Moderna mRNA-1273

mRNA-1273 vaccine Moderna mRNA-1273 (Moderna)

mRNA vaccination Messenger RNA vaccination

SARS-CoV S1-specific IgG Ab titers SARS-CoV-2 S1-specific (S1) IgG antibody titers

TBIL Total bilirubin

Tenofovir Tenofovir disoproxil

Additional Information and Declarations

Competing Interests

Author Contributions

Human Ethics

Data Availability

The authors declare that they have no competing interests.

Ryunjin Lee performed the experiments, prepared figures and/or tables, and approved the final draft.

Jiwan Choi performed the experiments, prepared figures and/or tables, and approved the final draft.

Eunkyeong Lee performed the experiments, prepared figures and/or tables, and approved the final draft.

Jooyoung Lee performed the experiments, prepared figures and/or tables, and approved the final draft.

Jiye Kim performed the experiments, prepared figures and/or tables, and approved the final draft.

Seoon Kang performed the experiments, prepared figures and/or tables, and approved the final draft.

Hye-In An performed the experiments, prepared figures and/or tables, and approved the final draft.

Sung-Han Kim analyzed the data, authored or reviewed drafts of the article, and approved the final draft.

Sung-Min Kim analyzed the data, authored or reviewed drafts of the article, and approved the final draft.

Eun-Kyoung Jwa analyzed the data, authored or reviewed drafts of the article, and approved the final draft.

Gil-Chun Park analyzed the data, authored or reviewed drafts of the article, and approved the final draft.

Jung-Man Namgoong analyzed the data, authored or reviewed drafts of the article, and approved the final draft.

Gi-Won Song analyzed the data, authored or reviewed drafts of the article, and approved the final draft.

Young-In Yoon conceived and designed the experiments, authored or reviewed drafts of the article, and approved the final draft.

Eunyoung Tak conceived and designed the experiments, authored or reviewed drafts of the article, and approved the final draft.

Sung-Gyu Lee analyzed the data, authored or reviewed drafts of the article, and approved the final draft.

The following information was supplied relating to ethical approvals (i.e., approving body and any reference numbers):

Institutional Review Board of Asan Medical Center (2021-0746).

The following information was supplied regarding data availability:

The raw data is available in the Supplemental File.

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
