# Peer review of "Effects of combined immunosuppressant and hepatitis B virus antiviral use on COVID-19 vaccination in recipients of living donor liver transplantation"

_PeerJ, doi:10.7717/peerj.18651_

## Round 0.1 · original submission · Major Revisions

Both reviewers have raised very pertinent points, especially how the abstract is not very focused as well as issues with methods and results. Addressing all these comments will help bring the manuscript to a more competitive standing.

Reviewer 1 ·

Basic reporting

The paper is written in fine language and style. The article structure is appropriate.The title is informative and relevant.
The references are relevant and recent. The cited sources are referenced correctly. Appropriate and key studies are included.
The introduction reveals what is already known about this topic. The research question also justified given what is already known about the topic.
All figures are with high quality and clear; raw data shared - clear and sufficient for analysis.

Experimental design

The research is within the scope of the journal.
The research question is clearly outlined. The aim is stated clear. The authors stated clearly what study found and how they did it.
The process of selection of the subjects was clear. The variables are well defined and measured appropriately. The study methods are valid and reliable. There are enough details provided in order to replicate the study.

Validity of the findings

The data is presented in an appropriate way. The text in the results add to the data and it is not repetitive. Statistically significant results are clear. It is clear which results are with practical meaning. Results are discussed from different angles and placed into context without being overinterpreted.
The conclusions answer the aim of the study. The conclusions are supported by references and own results.

Additional comments

Major issues - none
Minor issues
1. The graphical abstract - the left part of the figure is not clear - immunosuppressants and anti-HBV drugs are mentioned along with SARS-CoV-2, but the first two are attributions to the patient, and the virus - risk factor. Please, clarify.
2. The abstract - the background is too vague and not particularly related to the aim, please, focus on the vaccines for COVID-19 vaccination in recipients of Living Donor Liver Transplantation, but not the COVID-19 vaccination generally.

Reviewer 2 ·

Basic reporting

Thank you for the opportunity to review the manuscript regarding the efficacy of vaccines in preventing COVID-19 among liver transplant (LT) participants, including those on antiviral agents for hepatitis B virus. This study provides valuable data on the immunogenicity of vaccines in this vulnerable population. However, I have a few comments that could enhance the quality of the manuscript:
Introduction
• Line 81: Is it BNT162b1 or BNT162b2?
Methods
• The timeline for blood sampling and vaccination is unclear (lines 114-117). It would be helpful to specify the vaccination schedule for each vaccine, e.g., BNT162b2—three injections at 0, 1, and 6 months.
• Please provide the clinical trial registry number.
• Could you clarify why the authors used the cutoff values of mean + 3SD?
Results
• For how long were liver function tests (LFT) monitored after vaccination? For example, was it one month post-vaccination?
• It may not be accurate to combine the immunogenicity results of adenoviral vector-based vaccines and mRNA-based vaccines. A subgroup analysis for each vaccine type would be more appropriate.
• Given that most living donor liver transplant (LDLT) recipients received BNT162b2 (n=60), while most healthy controls received chAdOx1 (n=84), the immunologic results should not be merged.
• Figure 1: There is no control group indicated in this flow diagram.
• Table 1: Please include the type and numbers of immunosuppressants used by the participants.
Thank you for considering these suggestions to improve the manuscript.

Experimental design

Appropriate

Validity of the findings

Appropriate

Additional comments

As the attached file

Annotated reviews are not available for download in order to protect the identity of reviewers who chose to remain anonymous.

---

## Round 0.2 · accepted · Accept

The reviewers' comments have been sufficiently addressed.

Reviewer 1 ·

Basic reporting

The authors revised the manuscript based on the reviewers` comments and suggestions. The paper has been improved significantly.

Experimental design

No issues here.

Validity of the findings

The revision of the presentation of the results improved the paper's quality.

Additional comments

No additional comments.